# Does Physical Activity Have an Impact on Recurrence Dynamics in Early Breast Cancer Patients?

**DOI:** 10.3390/jcm10040831

**Published:** 2021-02-18

**Authors:** Elia Biganzoli, Christine Desmedt, Romano Demicheli

**Affiliations:** 1Laboratory of Medical Statistics, Biometry and Epidemiology “Giulio A. Maccacaro”, Department of Clinical Sciences and Community Health & DSRC, University of Milan, Campus Cascina Rosa, Fondazione IRCCS Istituto Nazionale Tumori, Via A. Vanzetti 5, 20133 Milano, Italy; romano.demicheli@guest.unimi.it; 2Laboratory for Translational Breast Cancer Research, Department of Oncology, KU Leuven, 3000 Leuven, Belgium; christine.desmedt@kuleuven.be

**Keywords:** breast cancer prognosis, pragmatic exercise, tumor dormancy, late metastases

## Abstract

Several studies have suggested that pre and/or postdiagnosis physical activity can reduce the risk of recurrence in breast cancer patients, however its effect according to follow-up time has not yet been investigated. We analyzed recurrence and mortality dynamics in randomized clinical trials (RCTs) from Australia and Canada. The combined Australian RCTs evaluated, at a median follow-up of 8.3 years, an 8-month pragmatic exercise intervention in 337 women with newly diagnosed breast cancer, while the Canadian RCT evaluated, at a median follow-up of 7.4 years, supervised aerobic or resistance exercise during chemotherapy in 242 patients. For each RCT, the control arm consisted of patients undergoing usual care. We estimated the event dynamics by the discrete hazard function, through flexible regression of yearly conditional event probabilities with generalized additive models. In the considered RCTs, the recurrence and mortality risk of patients enrolled in the physical activity arm were stably decreased at medium/long term after five year of follow-up. In the Australian RCTs where patients were recruited by urban versus rural area, the latter group did not display benefit from physical activity. Estimated odds ratios (95% confidence intervals) for disease-free survival (DFS) in urban women were 0.63 (0.22–1.85); 0.27 (0.079–0.90); 0.11 (0.013–0.96) at the 3rd, 5th and 7th year of follow-up, respectively. For rural women, DFS patterns were overlapping with odds ratios (ORs), approximating 1 at the different years of follow-up. Although not reaching statistical evidence, the estimates in the Canadian trial were in line with the results from the Australian urban women with ORs (95% CI) for DFS of 0.70 (0.33–1.50); 0.47 (0.19–1.18); 0.32 (0.077–1.29) at 3rd, 5th, 7th follow-up year, respectively. While we acknowledge that the analyzed RCTs were not designed for investigating disease recurrence over time, these results support the evidence that physical activity reduces the risk of developing medium-/long-term metastases. Additional translational research is needed to clarify the mechanisms underlying these observations.

## 1. Introduction

The concept that physical activity results in a substantial decrease of breast cancer risk relies on a wide number of observational epidemiological investigations. In a review of 73 studies by Lynch et al. [1], this reduction was estimated to be about 25% in physically active women as compared to the less active ones. Therefore, in spite of the observational nature of most studies on one hand, and, the heterogeneity of exercise prescription and quantitative assessment on the other hand, physical activity is an acknowledged preventive factor for breast cancer development. On the contrary, the prognostic effect of physical exercise in breast cancer patients is less evident. A meta-analysis by Lahart et al. [2] pointed out significant associations between recent pre-diagnosis physical activity and risk of breast cancer-related death. Moreover, post-diagnosis physical activity was associated to risk reduction of both all-cause and breast cancer-related death, although with evidence of heterogeneity across the studies. The review concluded that there is a need for randomized clinical trials (RCTs) investigating the role of physical activity on all-cause death and breast cancer outcomes. These results were further supported by recent evidence from the MARIE study, a population-based study of 3813 postmenopausal breast cancer patients [3]. Here, the authors showed that patients who are physically active have a better breast cancer prognosis than those who are inactive, especially for those who were not active before breast cancer diagnosis. Finally, a recent metanalysis investigating the pre and postdiagnosis physical activity on survival outcomes for different cancer types, showed that for breast cancer patients, there was a greater reduction in cancer-specific and all-cause mortality for postdiagnosis as compared to pre-diagnosis physical activity [4]. Of note, physical activity in the period following surgery, during and after adjuvant chemotherapy, if any, is associated with psychological benefits, especially when a therapeutic alliance is created between the patient and the technical operator who supervises this activity [5].

In breast cancer, the recurrence dynamics are characterized by a multipeak pattern [6], a pattern that may be well understood in the context of tumor dormancy, tumor homeostasis and acceleration of metastasis development after primary breast cancer surgery [6,7]. So far, the changes associated with physical activity have not been investigated according to follow-up time. According to the models of tumor dormancy, this information could support a link with specific phases of subclinical metastasis development. This knowledge is important for investigations aiming at elucidating biological mechanisms underlying the effectiveness of physical exercise [8]. In the present study, we re-analyzed data from RCTs [9,10] to explore the effects of physical activity on the dynamics of breast cancer recurrence and mortality according to follow-up time.

## 2. Materials and Methods

### 2.1. Patient Series

In this comparison between the recurrence dynamics between patients undergoing or not subjected to planned physical activity, we limited the analysis to RCTs where the timing of reported events was reliably assessable. In our survey of published clinical studies, only two reports met the inclusion characteristics [9,10].

### 2.2. Statistical Methods

To estimate the event dynamics according to follow-up time, we analyzed the published data from the above-mentioned RCTs by estimating the hazard function. This is the instantaneous event rate throughout the follow-up time, which, in a discrete framework, is represented by the conditional event probability at different time intervals, given survival before that time. From the report of Hayes [9], the hazard function was estimated directly from the numbers of patients reported in the figures, since there was no loss to follow-up. In contrast, in the Courneya report [10], the estimation of the hazard rates was performed by assessing both patients at risk and events in a given time interval from Kaplan–Meyer or cumulative incidences curves. A discretization of the time axis in yearly units was applied and a flexible regression of discrete hazards (estimated conditional event probabilities) [11] through generalized additive models was adopted [12]. Log odds were directly estimated from the model, given the canonical logit link function. Since the low discrete hazard values, estimated odds ratios (OR), are expected to closely approximate hazard ratios. Approximate confidence intervals (CI) for the ORs were computed from standard errors on model linear predictors followed by inverse logit transformation. All analyses were done using R software v. 3.6.1 (R Core Team (2020). R: A language and environment for statistical computing. R Foundation for Statistical Computing, Vienna, Austria).

## 3. Results

Hayes et al. [9] reported an analysis on two Australian RCTs according to the residence, designed to evaluate the protective effect of an 8-month pragmatic exercise intervention in 337 women with newly diagnosed breast cancer. Actually, 194 women were analyzed in the urban trial (*n* = 134 exercise and *n* = 60 usual care) and 143 women in the rural trial (*n* = 73 exercise and *n* = 70 usual care). Cox proportional hazards models were used to estimate hazard ratios (HRs) and 95% confidence intervals (CI) in single trials and in the pooled series. Considering the pooled series, they found that, after a median follow-up of 8.3 years, physical exercise was associated with better overall survival (OS) (HR = 0.45, 95% CI 0.20–0.96; *p* = 0.04) and disease-free survival (DFS) (HR = 0.66, 95% CI 0.38–1.17; *p* = 0.16). The prognostic improvement was only evident in the urban series and not in the rural one. Although they did not analyze recurrence and mortality dynamics, they reported cumulative incidence curves for DFS and OS for pooled, urban and rural breast cancer patients, randomized to the usual care or exercise arms, and listed the number of patients at risk in each follow-up year.

Courneya et al. [10] reported an exploratory analysis of data from the Supervised Trial of Aerobic versus Resistance Training (START), a Canadian multicenter trial that randomized 242 early breast cancer patients to usual care (*n* = 82) or exercise (*n* = 160) during chemotherapy. After a median follow-up of 89 months, the eight-year DFS favored the exercise groups (HR, 0.68; 95% (CI), 0.37–1.24; *p* = 0.21), as well as OS (HR, 0.60; 95% CI, 0.27–1.33; *p* = 0.21), distant DFS (HR, 0.62; 95% CI, 0.32–1.19; *p* = 0.15), and recurrence-free interval (RFI) (HR, 0.58; 95% CI, 0.30–1.11; *p* = 0.095). No analysis of event dynamics was carried out. However, similarly to Hayes, Kaplan–Meyer estimated DFS, distant DFS or cumulative incidences were reported for all efficacy endpoints as well as the number of patients at risk at each follow-up interval.

The Australian data [9] were analyzed per trial, (Figure 1) comparing the impact of physical activity within the urban (Figure 1a,b) and rural (Figure 1c,d) patients, both for DFS (Figure 1a–c) and OS (Figure 1b–d). Different hazard rate patterns were present between urban and rural women in the usual care group. By contrast, the hazard rate patterns from women in the exercise group were more similar. This configuration was coherently observed for both DFS and OS. In urban women, the recurrence and mortality reduction associated with physical activity cover late events occurring after five years of follow-up. To quantify this effect, we estimated the ORs (95% CI) for DFS in urban women, which were 0.63 (0.22–1.85); 0.27 (0.079–0.90); 0.11 (0.013–0.96) at the 3rd, 5th, 7th follow-up year, respectively. This finding shows the increasing protective trend with statistical evidence from the 4th to 5th year. Concerning OS, we observed similar results in the urban population with an average OR of 0.27, although without statistical evidence. For rural women, overlapping DFS patterns revealed no benefit from exercise with ORs approximating 1, while estimated ORs for OS were stably halved following exercise although without statistical evidence. Figure 2 reports the hazard rate patterns resulting from the analysis of the Canadian trial [10]. The hazard rate patterns for these Canadian patients were very similar to the patterns of the Australian urban women, both for those enrolled in the exercise group and for those undergoing usual care. The similarity involves the timing of exercise-related improvement that emerges in the later follow-up years. Here, the estimated ORs (95% CI) for DDFS were 0.70 (0.33–1.50); 0.47 (0.19–1.18); 0.32 (0.077–1.29) at 3rd, 5th, 7th follow-up year, respectively, and values for OS were comparable. Interestingly, although not reaching statistical evidence, the estimated ORs and trends were comparable to the corresponding results for the Australian trial in urban women.

## 4. Discussion

In this study, we re-analyzed data from RCTs that compared postdiagnosis physical exercise to usual care in early breast cancer patients [9,10]. Our aim was to identify whether physical exercise would influence recurrence dynamics, i.e., the timing of tumor recurrence over follow-up. This can be of interest since the lag time between primary tumor removal and tumor recurrence is related to the state of microscopic metastatic foci at the time of surgery and to their growth pattern. Therefore, the recurrence dynamics, which may be suitably described by the hazard function, could provide insights on the biological behavior of metastases. For breast cancer, this analytical approach already provided meaningful results in the past [6,7].

Although neither the Australian nor the Canadian trials were designed and powered for analyses of survival outcomes, they provide an opportunity to conduct hypotheses-generating analyses, in the context of randomized comparisons. The considered RCTs support the conclusion that postdiagnosis exercise during and beyond breast cancer treatment may improve breast cancer outcomes, at least in certain subsets of patients. In the present report, we are adding the notion that physical exercise, when effective, is apparently correlated to the reduction of clinical recurrence and mortality hazards in later years following primary treatment (about 5 years).

Of note, the relevant measured effects averaged over time were similar with the results of the recent metanalysis by Friedenreich et al. [4].

These results could be interpretable in the context the above recalled model [6,7] according to which primary surgical removal results in the sudden wake-up of subclinical dormant metastases that originate the multipeak pattern of the hazard rate for DM observed during the follow-up. Therefore, our findings draw attention to specific phases of metastasis development and may orient research aimed at discovering the mechanisms underlying the effects of physical activity.

Additional aspects related to the beneficial effect of physical activity should also be considered. For instance, an immunology-based mechanism is suggested by investigations providing evidence that skeletal muscle behaves as an endocrine organ, capable of expressing and secreting cytokines into the circulatory system during physical activity, a finding that connects active skeletal muscles to the maintenance of a healthy immune system during ageing [13]. Moreover, physical activity has already shown to reduce levels of inflammatory markers in circulation, such as IL-6, in the Yale Exercise and Survivorship Study [14]. A recent experimental study also suggested a new mechanism in which exercise reduces inflammatory cell production via the reduction of leptin levels and instruction of the hematopoietic progenitor cells [15]. Therefore, on the basis of the review propositions [13], and following our previous research [16,17], we hypothesize that while short term pharmacological interventions such as adjuvant chemotherapy may act mainly on early recurrences [6] (possibly associated to surgery-related acute inflammatory processes), physical exercise might deal with later recurrences through its modulatory effects on immune processes, possibly related to chronic inflammation. Actually, breast cancer recurrence is associated with adiposity [16], so exercising should play a relevant role as a component to a weight control program with a series of side benefits in aftercare rehabilitation.

Our analysis has a few obvious limitations. Firstly, the two Australian trials, the rural and urban one, did not display fully consistent event dynamics, in spite of similar disease prognostic factors and exercise interventions [9]. This inconsistency suggests possible residential area-related factors not accounted for. This hypothesis is in part supported by the similarity of event dynamics between the urban Australian and the Canadian trials that involved women from three metropolitan areas [9,10]. Secondly, the definition of outcomes presented differences between the Australian and Canadian studies. For instance, the definition of DFS considered by Hayes et al. [9] included breast cancer recurrence, new primary breast cancer or death from any cause, while the most similar outcome from Courneya et al. [10], distant DFS, included distant recurrence, second primary invasive cancer and death from any cause. In spite of the lack of complete consistency between assessed outcomes, the similarity of event dynamics was nevertheless evident as well as the trends in the protective effects of physical activity according to ORs. Lastly, our analysis was performed by extracting data from published figures instead of processing original data from single patients. However, according to our past experience on hazard rate assessment, no major drawbacks should be expected.

## 5. Conclusions

This re-analysis of RCTs investigating the effect of physical activity on disease recurrence dynamics provides evidence that physical activity reduces the risk of developing medium and long-term events. These findings should set the scene for further investigations on the effect of physical exercise in the context of tumor dormancy and surgery-related metastasis development.

## Figures and Tables

**Figure 1 jcm-10-00831-f001:**
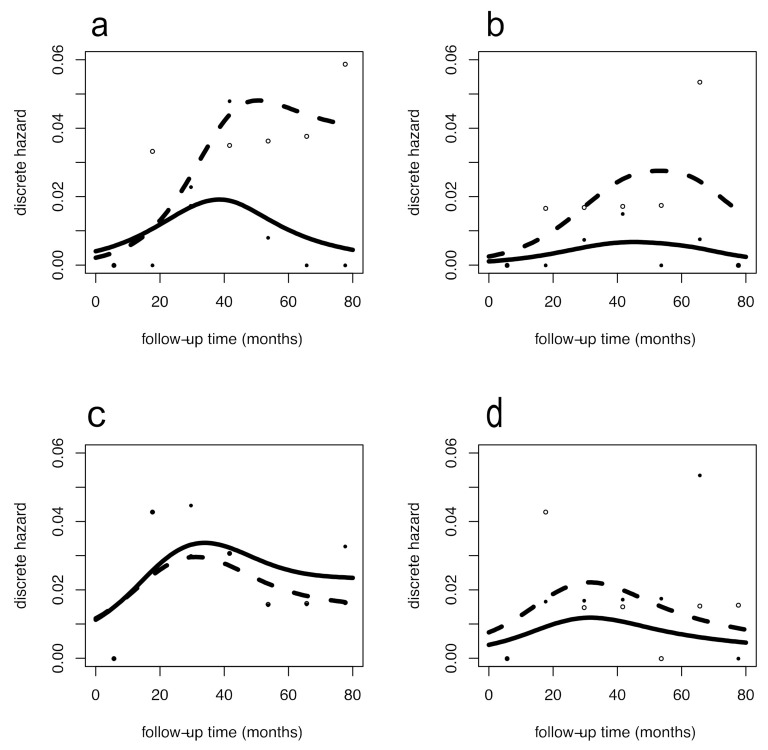
Recurrence and overall mortality dynamics in the urban and rural Australian trials. Hazard rate patterns for disease-free survival (DFS) (left panels: (**a**,**c**)) and odds ratios (OS) (right panels: (**b**,**d**)) for 194 urban women (upper panels: (**a**,**b**)) and 143 rural women (bottom panels: (**c**,**d**)). Discrete cause-specific hazard rates were estimated within a yearly interval, empirical estimates are represented as small open circles for no exercise and filled circles for exercise. Smoothed curves were obtained by flexible regression procedure based on generalized additive models, dashed lines for no exercise and continuous lines for exercise. *X* axis: time in months; *y* axis: discrete hazard rates (estimated annual conditional event probabilities).

**Figure 2 jcm-10-00831-f002:**
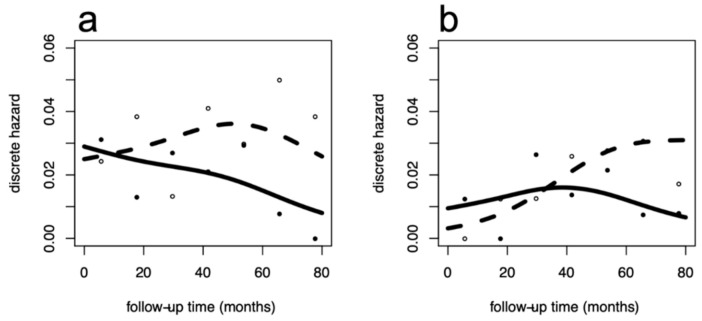
Recurrence and overall mortality dynamics in the Canadian trial. Hazard rate patterns for Distant DFS (left panel: (**a**)) and OS (right panel: (**b**)) in the 242 patients considered in this trial. Discrete cause-specific hazard rates were estimated within a yearly interval, empirical estimates are represented as small open circles for no exercise and filled circles for exercise. Smoothed curves were obtained by flexible regression procedure based on generalized additive models, dashed lines for no exercise and continuous lines for exercise. *X* axis: time in months; *y* axis: discrete hazard rates (estimated yearly conditional event probabilities).

## Data Availability

Data sharing not applicable to this article as no datasets were generated during the current study.

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
