# Peer review of "Does Physical Activity Have an Impact on Recurrence Dynamics in Early Breast Cancer Patients?"

_jcm, 2021, doi:10.3390/jcm10040831_

Round 1
Reviewer 1 Report
The article is of scientific interest and in line with the aims of the journal. The authors guidelines have been respected and the article does not require a native speaker to review the English language.
The topics covered are in accordance with the methodology followed and well represented. There are only a few concerns to correct in my opinion.
TITLE
The title creates an expectation in the reader that is somehow betrayed. Specify the type of study and refer to the 2 included studies (canada and australia). for example "Does physical activity have an impact on recurrence dynamics in early breast cancer patients?"
KEYWORD
In order to increase the visibility of the article, do not use keywords already present in the title.
INTRODUCTION
In the introduction, mention the psychological side that is involved in compliance with physical exercise and use reference:
- Paolucci T, Bernetti A, Paoloni M, Capobianco SV, Bai AV, Lai C, Pierro L, Rotundi M, Damiani C, Santilli V, Agostini F, Mangone M. Therapeutic Alliance in a Single Versus Group Rehabilitative Setting After Breast Cancer Surgery : Psychological Profile and Performance Rehabilitation. Biores Open Access. 2019 Jul 3; 8 (1): 101-110. doi: 10.1089 / biores.2019.0011. PMID: 31275735; PMCID: PMC6607049.
MATERIALS AND METHODS
The inclusion and exclusion criteria should be specified in the materials and methods. Why did you only include 2 studies? Why have you excluded the others?
The summary of the studies should be included in the first part of the results section.
RESULTS AND DISCUSSION
The results are written very fluently and well discussed in the discussion section.
REFERENCES
The references are relevant to the topic and recent. Add suggested reference.
FIGURES
The figures are of good quality and adequately complement the text.
Author Response
TITLE
The title creates an expectation in the reader that is somehow betrayed. Specify the type of study and refer to the 2 included studies (canada and australia). for example "Does physical activity have an impact on recurrence dynamics in early breast cancer patients?"
R: The Title was changed as requested
KEYWORD
In order to increase the visibility of the article, do not use keywords already present in the title.
R: The new keywords were modified as follows: breast cancer prognosis, pragmatic exercise, tumor dormancy, late metastases
INTRODUCTION
In the introduction, mention the psychological side that is involved in compliance with physical exercise and use reference:
- Paolucci T, Bernetti A, Paoloni M, Capobianco SV, Bai AV, Lai C, Pierro L, Rotundi M, Damiani C, Santilli V, Agostini F, Mangone M. Therapeutic Alliance in a Single Versus Group Rehabilitative Setting After Breast Cancer Surgery : Psychological Profile and Performance Rehabilitation. Biores Open Access. 2019 Jul 3; 8 (1): 101-110. doi: 10.1089 / biores.2019.0011. PMID: 31275735; PMCID: PMC6607049.
R: As requested, it was added a specific sentence with the due reference.
Of note, physical activity in the period following surgery, during and after adjuvant chemotherapy, if any, is associated with psychological benefits, especially when a therapeutic alliance is created between the patient and the technical operator who supervises this activity [5].
MATERIALS AND METHODS
The inclusion and exclusion criteria should be specified in the materials and methods. Why did you only include 2 studies? Why have you excluded the others?
R: The inclusion and exclusion criteria were added.
In this comparison between the recurrence dynamics between patients undergoing or not subjected to planned physical activity, we limited the analysis to RCTs where the timing of reported events was reliably assessable. In our survey of published clinical studies only two reports met the inclusion characteristics [9,10]
The summary of the studies should be included in the first part of the results section.
R: Summaries were moved to the Results section.
RESULTS AND DISCUSSION
The results are written very fluently and well discussed in the discussion section.
REFERENCES
The references are relevant to the topic and recent. Add suggested reference.
R: The suggested reference was added.
FIGURES
The figures are of good quality and adequately complement the text.
Reviewer 2 Report
In the manuscript 'Impact of physical activity on recurrence dynamics in early breast cancer patients' authors try to highlight very important question. I have few comments as below-
- what is the role of aftercare in these kind of studies, does this impact the outcome?
- Heterogeneity plays a critical role in these kind of studies, what are authors thought about heterogeneity and adverse effect. Are there any studies suggested so?
Author Response
1. what is the role of aftercare in these kind of studies, does this impact the outcome?
R: We thank the Reviewer for this consideration. We added the following statement in the discussion:
“Actually, breast cancer recurrence is associated with adiposity [16] so exercising should play a relevant role as a component to a weight control program with a series of side benefits in aftercare rehabilitation.“
2. Heterogeneity plays a critical role in these kind of studies, what are authors thought about heterogeneity and adverse effect. Are there any studies suggested so?
R: We agree with the Reviewer about the role of heterogeneity in this kind of studies. However, a formal assessment of its effect was outside the scope of the present work analyzing the results of few clinical trials in specific healthcare settings. We are not aware of suitable relevant studies.